# Anti-Inflammatory Activity and Mechanism of Sweet Corn Extract on Il-1β-Induced Inflammation in a Human Retinal Pigment Epithelial Cell Line (ARPE-19)

**DOI:** 10.3390/ijms24032462

**Published:** 2023-01-27

**Authors:** Inthra Koraneeyakijkulchai, Rianthong Phumsuay, Parunya Thiyajai, Siriporn Tuntipopipat, Chawanphat Muangnoi

**Affiliations:** 1Master of Science Program in Nutrition, Faculty of Medicine, Ramathibodi Hospital and Institute of Nutrition, Mahidol University, Nakhon Pathom 73170, Thailand; 2Cell and Animal Model Unit, Institute of Nutrition, Mahidol University, Nakhon Pathom 73170, Thailand; 3Food Chemistry Unit, Institute of Nutrition, Mahidol University, Nakhon Pathom 73170, Thailand

**Keywords:** sweet corn, age-related macular degeneration, anti-inflammatory activity, human retinal pigment epithelial cell line, carotenoids

## Abstract

Age-related macular degeneration (AMD) is an eye disease associated with aging. Development of AMD is related to degeneration and dysfunction of the retinal pigment epithelium (RPE) caused by low-grade chronic inflammation in aged RPE cells leading to visual loss and blindness. Sweet corn is a good source of lutein and zeaxanthin, which were reported to exert various biological activities, including anti-inflammatory activity. The present study aims to investigate the anti-inflammatory activity and mechanisms of SCE to inhibit the production of inflammatory biomarkers related to AMD development. Cells were pretreated with SCE for 1 h followed by stimulation with IL-1β for another 24 h. The results demonstrated that SCE attenuated IL-1β-induced production of IL-6, IL-8, and MCP-1 and the expression of ICAM-1 and iNOS in a dose-dependent manner. In addition, SCE suppressed the phosphorylation of ERK1/2, SAPK/JNK, p38, and NF-κB (p65) in IL-1β-stimulated ARPE-19 cells. These results proved that SCE protected ARPE-19 cells from IL-1β-induced inflammation by inhibiting inflammatory markers partly via suppressing the activation of MAPK and NF-κB signaling pathways. Overall, SCE is a potential agent for the prevention of AMD development, which should be further evaluated in animals.

## 1. Introduction

Age-related macular degeneration or AMD is a disease of the eye characterized by degeneration of the macula in the retina, which is a major cause of vision loss and irreversible blindness in people in the developing world, especially elderly people [1]. As it progresses, AMD ultimately causes vision loss [2]. An epidemiological study reported that 5.4% of blindness in adults over the age of 50 is caused by the development of macular degeneration [3]. In addition, the incidence of AMD will continuously and rapidly increase due to the increase in average life expectancy and the exponential growth of the aging population [4]. A meta-analysis and systemic review reported that the prevalence of AMD is increasing, and estimated 288 million cases in 2040 because of the increase in the aging population [5]. Clinically, AMD is divided into two main phenotypes, dry AMD (geographic atrophy, non-neovascular, or non-exudative) and wet AMD (neovascular or exudative) [6]. The wet form of AMD can be typically cured with medicine (anti-vascular endothelial growth factor (anti-VEGF)), injected into the eye to prevent blood vessel growth. However, there is neither a preventive therapy nor an effective treatment for the dry form of AMD, which is regarded as a global eye health problem causing a substantial economic and social burden [7]. Thus, discovering effective treatments for preventing or delaying AMD is urgent.

Although the development of AMD is complex, the major causes of AMD progression and development are inflammation and oxidative stress [8]. Chronic inflammation is the cause of formation and acceleration of drusen which are a maker of early AMD. Therefore, various inflammatory markers were also proposed as crucial biomarkers for AMD [9]. Retinal pigment epithelium (RPE) degeneration is a major hallmark of AMD [10]. The RPE, located between the photoreceptors and Bruch’s membrane, is a monolayer of cuboidal cells containing many melanosomes with pigmented color joined laterally with their apices by tight junctions (TJs) [11]. The main functions of the RPE are to promote and maintain retinal health, including the transport of fluid and nutrients from the choroid and removal of waste; phagocytosis, and digestion of photoreceptors’ worn-out outer segments (photoreceptor renewal); visual cycle involvement; and establishment of the blood–retinal barrier (BRB) [12]. In addition, the RPE is also involved in regulating immune and inflammatory responses to stress in the retina [13].

RPE cells have a high metabolic rate and exist in a reactive oxygen species (ROS)-rich environment. Therefore, the accumulation of oxidative damage in cellular components, including proteins, lipids, DNA, and mitochondria, occurs during the normal aging of RPE cells. This leads to the generation of oxidation-specific epitopes such as malonaldehyde (MDA), 4-hydroxy-2-nonenal (HNE), advanced glycation end products (AGEs), and carboxyethyl pyrrole (CEP) [14]. It is well known that mitochondrial damage and ROS production induce inflammation in RPE cells through the production of pro-inflammatory mediators, such as cytokines (e.g., interleukin-1β (IL-1β), and interleukin-6 (IL-6)), and chemokines (e.g., monocyte chemoattractant protein-1 (MCP-1) and interleukin-8 (IL-8)); mediators of adherence of leukocytes (intercellular adhesion molecule-1 (ICAM-1)); and pro-inflammatory enzymes (inducible nitric oxide synthase (iNOS)) [15,16]. Previous studies showed that the levels of inflammatory mediators, including IL-6, IL-8, MCP-1, and ICAM-1 in serum or ocular tissue, are changed in exudative AMD development and progression [17].

IL-1β, a key pro-inflammatory cytokine, promotes the upregulation of chemokines and initiates an innate immune response associated with inflammation, infection, and autoimmunity [18]. It is one of the most potent cellular stimulators for immune cells and other cell types, thereby mediating multiple immune responses during ocular inflammation [19]. The inflammation process is modulated via mitogen-activated protein kinase (MAPK) signaling pathways, including the extracellular signal-regulated kinase (ERK), c-Jun N-terminal kinase (JNK), and p38 MAPK and nuclear factor kappa-B (NF-κB) pathways; all of which lead to an elevation in the production of cytokines and chemokines [20,21]. These, in turn, promote macrophage infiltration and polarization for a prolonged period, resulting in the release of endogenous molecules that stimulate the immune system including innate and acquired immunity, thereby damaging RPE cells in the retina and promoting the development of AMD [8,14]. Therefore, pharmacological agents targeting inflammation in RPE could be an appealing strategy for treating AMD.

Corn, especially sweet corn (*Zea mays* L. ssp. saccharata Sturt), is a monocotyledon of the family Gramineae (Poaceae) and is native to North America. It is a highly favored variety due to its sweet taste. Previous studies have shown that sweet corn contains various nutrients and phytochemicals, such as carotenoids [22]. It is a rich and good source of lutein and zeaxanthin, which can exert anti-inflammatory activity to alleviate inflammation in RPE cells [22,23]. Several studies have shown the protective effect of lutein and zeaxanthin to reduce the risk of AMD by suppressing the production of inflammatory mediators such as IL-6, IL-8, MCP-1, ICAM-1, and iNOS through the deactivation of MAPK and NF-κB signaling pathways in vitro and in vivo [24,25,26]. Previous studies assessed the protective effects of pure lutein and zeaxanthin against inflammation in activated human retinal pigment epithelial cells (ARPE-19) and other cell types. However, to date, no study examined the effect of sweet corn extract, a dietary extract rich in carotenoids, against IL-1β-induced inflammation in ARPE-19 cells. ARPE-19 is a human cell line exhibiting structural features and functional properties similar to primary human RPE cells [27]. Hence, this is the first study to report the protective effects of sweet corn extract on inflammation in retinal pigment epithelial cells.

## 2. Results

### 2.1. Sweet Corn Extract Profiled Using HPLC–DAD

High-performance liquid chromatography coupled with diode array detection (HPLC–DAD) analysis of sweet corn extract (SCE) extracted from mixed solvent was performed to identify the main carotenoids in SCE. The amount of each carotenoid (lutein, zeaxanthin, β-cryptoxanthin, α-carotene, and β-carotene) in SCE is shown in Table 1. We confirmed that the highest peak identified was zeaxanthin, followed by lutein, and minor amounts (less than 10%) of β-carotene, α-carotene, and β-cryptoxanthin (Figure 1).

### 2.2. Effect of SCE on Cell Viability of ARPE-19 Cells

We tested different concentrations of SCE ranging from 1 to 100 μg/mL on ARPE-19 cells to determine non-toxic concentrations for further studies. An MTT assay was used to determine the cytotoxicity of SCE on ARPE-19 cells. Results showed that ARPE-19 cells treated with SCE up to 100 μg/mL for 24 h did not experience cytotoxic effects in comparison to the control group (Figure 2A). Inverted microscopy images also supported the lack of cytotoxic effects of these concentrations of SCE on the viability of ARPE-19 cells (Figure 2B). Thus, the highest non-toxic concentration of SCE at 100 µg/mL was used for subsequent experiments to evaluate its protective effects on ARPE-19 cells.

### 2.3. Effect of IL-1β on IL-6 and IL-8 Secretion and ARPE-19 Cell Viability

We investigated concentrations of IL-1β ranging from 0.1 to 10 ng/mL in ARPE-19 cells to select the lowest concentration of IL-1β that induces inflammation in ARPE-19 cells without toxicity. The results showed that ARPE-19 cells treated with IL-1β at a concentration of 0.1–10 ng/mL for 24 h significantly increased IL-6 and IL-8 secretion compared to the control group (*p* < 0.05) without cytotoxicity (Figure 3A–C). In subsequent experiments, the lowest tested concentration of 0.1 ng/mL IL-1β was used to induce inflammation.

### 2.4. Effect of SCE on IL-6, IL-8, and MCP-1 Secretion in ARPE-19 Cells Induced with IL-1β

We further evaluated the protective effect of SCE in three serial dilutions on the expression of inflammatory markers (IL-6, IL-8, and MCP-1) in IL-1β-induced ARPE-19 cells. Cells were treated with SCE at 1, 10, and 100 µg/mL for 1 h before induction with IL-1β at 0.1 ng/mL for 24 h. Cell culture media were collected to measure IL-6, IL-8, and MCP-1 concentration using ELISA. IL-1β significantly stimulated ARPE-19 cells to secrete IL-6, IL-8, and MCP-1 compared to the control group (*p* < 0.05). The results indicated that pretreatment with SCE could decrease the secretion of IL-6, IL-8, and MCP-1 from ARPE-19 cells induced with IL-1β in a dose-dependent manner (Figure 4A–C) without cytotoxicity (Figure 4D). SCE alone at a concentration of 100 µg/mL did not significantly affect IL-1β-induced IL-6, IL-8, and MCP-1 secretion in ARPE-19 cells.

### 2.5. Effect of SCE on iNOS and ICAM-1 Expression in ARPE-19 Cells Induced with IL-1β

ARPE-19 cells were treated as described in 2.4. After 24 h induction with IL-1β, cell lysates were collected to measure iNOS and ICAM-1 expression using immunoblot. The results shown in Figure 5 showed that IL-1β significantly stimulated the expression of iNOS and ICAM-1 protein in ARPE-19 compared to the control group (*p* < 0.05). SCE at concentrations of 1, 10, and 100 µg/mL significantly decreased iNOS and ICAM-1 expression compared to the IL-1β group (*p* < 0.05) in a dose-dependent manner. ARPE-19 cells treated with 100 µg/mL SCE alone did not show significant changes in iNOS and ICAM-1 expression. These results indicated that SCE at concentrations of 1, 10, and 100 µg/mL could attenuate the expression of iNOS and ICAM-1 from IL-1β-induced ARPE-19 cells in a dose-dependent manner without toxicity.

### 2.6. Effect of SCE on the MAPK Signaling Pathway in ARPE-19 Cells Induced with IL-1β

We further investigated the effect of SCE on MAPKs, such as ERK1/2, SAPK/JNK, and p38 MAPK. ARPE-19 cells were incubated as described in 2.4. After 24 h induction with IL-1β, cell lysates were collected to measure MAPK protein expression using immunoblots. Results showed that IL-1β induced inflammation in ARPE-19 cells through the activation of MAPKs by up-regulating the expression of the phosphorylated form of p38, ERK1/2, and SAPK/JNK without alteration of total p38, ERK1/2, and SAPK/JNK (Figure 6A–C). SCE significantly decreased p-p38, and p-SAPK/JNK protein expression at 1–100 µg/mL compared with cells incubated with IL-1β alone (*p* < 0.05), whereas SCE led to a significantly decreased p-ERK1/2 expression at only 10 and 100 µg/mL (Figure 6A–C). ARPE-19 cells treated with 100 µg/mL SCE alone did not experience a significant effect on the expression of the phosphorylated form of p38, ERK1/2, and SAPK/JNK. These results indicated that SCE could suppress the expression of p-p38, p-SAPK/JNK, and p-ERK1/2 in ARPE-19 cells induced with IL-1β.

### 2.7. Effect of SCE on the NF-κB Signaling Pathway in ARPE-19 Cells Induced with IL-1β

Upon exposure of ARPE-19 cells to IL-1β for 24 h, phosphorylated forms of NF-κB p65 (p-NF-κB p65) significantly increased in comparison to the control group (*p* < 0.05), as shown in Figure 7. ARPE-19 cells treated with SCE prior to induction with IL-1β decreased p-NF-κB p65 expression in a dose-dependent manner relative to the control group. Treatment with 100 µg/mL SCE alone did not have a significant effect on the expression of p-NF-κB p65 in ARPE-19 cells. These results indicated that SCE could reduce the expression of p-NF-κB p65 in ARPE-19 cells induced with IL-1β.

## 3. Discussion

Inflammation is associated with the pathology and physiology of various ocular diseases, including AMD and diabetic retinopathy [28]. During an inflammatory response within the retina, RPE cells are a major source of secreted pro-inflammatory mediators including cytokines, chemokines, and adhesion molecules [29]. IL-1β, a pro-inflammatory cytokine, plays an early role in triggering inflammatory responses attracting more inflammatory cells to the retina and promoting impairment and degeneration of the retina [19,30]. Any approach to suppressing the production of inflammatory mediators by activated RPE cells has the potential to prevent or slow down retinal pathogenesis. Currently, there is no single treatment known to cure AMD. However, bioactive constituents in foods with potent anti-inflammatory effects may be another choice to reduce the risk for or mitigate the symptoms of wet and dry AMD. In addition, they are affordable and relatively safe compared to therapeutic drugs. Regarding ocular health, several studies indicated that dietary lutein and zeaxanthin (xanthophylls) have strong anti-inflammatory properties both in vivo and in vitro [24,25,26,31]. In addition, a high intake of xanthophyll carotenoid (lutein/zeaxanthin)-rich diets was found to reduce prevalent age-related macular degeneration in both early and late phases [32,33]. Yellow sweet corn is an excellent source of lutein and zeaxanthin [22]. Yellow sweet corn contains not only carotenoids but also other anti-inflammatory phytochemicals, including phenolic acids and flavonoids [22,34,35]. Therefore, here we performed a detailed exploration of the anti-inflammatory activity and mechanism of sweet corn extract (SCE) on IL-1β-stimulated ARPE19 cells.

In the present study, we initially determined the carotenoid profiles of SCE. We found that lutein and zeaxanthin were the main carotenoids with a minor content of β-carotene, α-carotene, and β-cryptoxanthin. The results are in agreement with data from previous studies reporting that the major identified carotenoids in sweet corn were lutein, zeaxanthin, and α-cryptoxanthin and small amounts of β-carotene, α-carotene, β-cryptoxanthin [36]. The SCE contained zeaxanthin as more than 50% of the total carotenoid content, and the ratios of lutein to zeaxanthin (L/Z) were 0.7, which is consistent with data from other studies presenting the ratio of these carotenoids (L/Z) in sweet corn (*Zea mays* L.) at 0.6 [37]. These results indicated that sweet corn is a good source of lutein and zeaxanthin, close to other sources such as pumpkin, broccoli, and pea [38]. When we compared the amounts of carotenoids in sweet corn to other studies, we could show that the sweet corn’s carotenoid content (lutein, zeaxanthin, β-carotene, α-carotene, and β-cryptoxanthin) in this study was higher compared to other studies using raw grains [22,38]. This is because the raw sweet corn in our study was subjected to a thermal process, i.e., boiling, before being prepared for freeze–drying of sweet corn powder and extraction. Cooking was found to be the most useful method to obtain the highest yield of carotenoids by helping in the breakdown and disintegration of carotenoid–protein complexes and other physical barriers present in the plant sample, compared to other methods [39]. Therefore, the sweet corn’s carotenoid content measured is significantly increased when the samples are cooked.

Several inflammatory mediators such as IL-6, IL-8, MCP-1, ICAM-1, and iNOS are major inflammatory markers in the inflammation process involved in the initiation and propagation of AMD [16]. The pro-inflammatory cytokine IL-6 is a key mediator implicated in geographic atrophy, angiogenesis, and subretinal fibrosis [40,41]. The chemokines IL-8 and MCP-1 not only initiate inflammatory responses but also have a potent proangiogenic ability, induce surface expression of adhesion molecules, and facilitate leukocyte transmigration into ocular tissue in AMD progression [42]. ICAM-1 is a cell surface glycoprotein involved in choroidal neovascularization and facilitating leukocyte transmigration into ocular tissue along with MCP-1 [43]. In addition, iNOS has also been reported to be involved in the progression of AMD. It is a vital factor responsible for VEGF-mediated choroidal neovascularization and is used as an indicator for predicting the pathogenesis of wet AMD [44]. Thus, the inhibition of IL-6, IL-8, MCP-1, ICAM-1, and iNOS might be a useful therapeutic approach to ameliorate the pathogenesis of chronic inflammatory diseases such as AMD.

In in vitro studies demonstrated that IL-1β activated RPE cells to secrete cytokines and chemokines [45]. A number of previous studies showed that IL-1β upregulated the expression of IL-6, IL-8, ICAM-1, MCP-1, and iNOS in human RPE cells [45,46]. These activities are modulated via the NF-κB and MAPK signaling pathways [45]. The present study demonstrated that IL-1β induced ARPE-19 cells to produce pro-inflammatory mediators, including IL-6, IL-8, MCP-1, ICAM-1, and iNOS, partly via activation of MAPK and NF-κB signaling pathways, which is consistent with the data from these previous reports. However, SCE rich in lutein and zeaxanthin suppressed IL-1β-induced IL-6, IL-8, and MCP-1 secretion and ICAM-1 and iNOS expression in a dose-dependent manner. Previous studies reported that lutein and zeaxanthin could be taken up by ARPE-19 cells better than β-carotene and lycopene [47,48]. This evidence indicated that ARPE-19 cells might express proteins that bind to xanthophyll, such as SR-B1, CD36, GSTP1, and IRBP, similar to human RPE [47,49]. Other previous studies also confirmed the anti-inflammatory effects of lutein and zeaxanthin on ARPE-19 [50]. Thus, regular consumption of sweet corn may decrease the pathogenesis of such diseases.

In the inflammatory response, the MAPK pathway is activated by several inducers and upregulates the expression of inflammatory mediators, including cytokines and chemokines. MAPK activation can modulate the severity of some inflammatory diseases, including AMD [51]. Several studies have confirmed that the activation of MAPKs is involved in the IL-1β-induced expression of inflammatory mediators in different cell types, including ARPE-19 cells [45]. The inhibition of the MAPK pathway in IL-1β-induced ARPE-19 cells decreased the expression and production of IL-6, IL-8, MCP-1, and ICAM-1 [45]. MAPK inhibitors, including an ERK1/2 inhibitor (U0126), a p38 inhibitor (SB202190), and a JNK inhibitor (SP600125), were used to investigate the role of these pathways in inflammation in IL-1β-induced ARPE-19 cells [45,52]. Administration of MAPK inhibitors decreased the expression of soluble ICAM-1 (sICAM-1), while U0126 and SB202190 inhibited the expression of IL-6, IL-8, and MCP-1, but SP600125 could not [45]. Such data confirm that MAPK pathways play a major role in the expression of inflammatory mediators. A review reported using specific MAPK inhibitors as potential therapeutic targets for the treatment of AMD [53]. The results in the present study provided evidence that SCE protected ARPE-19 cells from IL-1β-induced inflammation by inhibiting IL-6, IL-8, MCP-1, ICAM-1, and iNOS production. The inhibitory effect of SCE observed was partly due to its ability to inhibit phosphorylation of MAPK signaling pathways (p38, ERK1/2, and SAPK/JNK), thereby ameliorating the inflammatory response.

Several studies have shown that NF-κB transcription factor regulates inflammatory processes by controlling the gene expression and protein levels of cytokines and chemokines in various cell types [54,55]. The inflammatory signals generated by IL-1β stimulate ARPE-19 cells to upregulate phosphorylated IKKα/β and lead to IκB phosphorylation and stimulation of NF-κB by phosphorylation at several serine residues on p65, especially, Ser536 [56]. The activated NF-κB then moves into the nucleus, where it is phosphorylated, upregulating inflammatory gene expression. Knockdown of NF-κB was shown to reduce the expression of ICAM-1, sICAM-1, IL-6, IL-8, and MCP-1 in inflammation in ARPE-19 cells induced by IL-1β [45]. NF-κB family members are known to play a major role in the regulation of gene expression in chronic retinal degenerative disorders, including AMD. Several in vitro studies indicated that the phosphorylation of NF-κB correlated with inflammatory molecules in AMD pathogenesis [57,58]. A previous study observed that Bay11–7082 (an NF-κB inhibitor) decreased the expression of ICAM, sICAM-1, IL-6, IL-8, and MCP-1 in ARPE-19 cells after IL-1β-induced inflammation, which was involved in a decrease in NF-κB signaling protein expression [45]. Thus, these data confirmed that NF-κB is the upstream regulator for cytokine and chemokine production in this cell model. The present study provided evidence that SCE protected ARPE-19 cells from IL-1β-induced inflammation by decreasing IL-6, IL-8, MCP-1, ICAM-1, and iNOS production via inhibiting phosphorylation of NF-κB p65.

However, other bioactive compounds with anti-inflammatory activity in SCE, such as phenolic acid, might also exert anti-inflammatory effects. The antioxidant and anti-inflammatory activities of dietary phenolic compounds have been extensively demonstrated in vitro and in vivo [35]. Phenolic acid and phenolic compounds including ferulic acid (FA) and p-coumaric acid (p-CA) are present in sweet corn [22]. FA is the main phenolic acid found in maize grain or corn, followed by p-CA [59]. A study reported that FA could attenuate IL-6 production in LPS-induced ARPE-19 cells [60]. Recent studies showed that FA supplements markedly inhibited pathological angiogenesis of the retina both in vivo and in vitro [61]. They decreased the expression of IL-6 and TNF-α and the production of iNOS, while anti-inflammatory IL-10 was upregulated both in hypoxia-treated BV2 cells and a retinopathy mouse model through inhibition of the NF-κB pathway [61]. Moreover, FA orchestrated microglia or macrophage polarization from M1 (pro-inflammatory phenotype) to M2 (anti-inflammatory phenotype) [61]. In addition, FA could inhibit tube formation and migration of human retinal endothelial cells (HRECs), which supported that FA displayed a potent anti-angiogenic activity in hypoxia-induced angiogenesis [61].

Macrophages are a major cell type involved in chronic inflammation observed in AMD tissue including subretinal microglia and lymphocytes [62]. Anti-inflammatory effects of p-CA were reported in the RAW 264.7 mouse macrophage cell line in LPS-induced inflammation. Pretreatment with p-CA could inhibit the expression of iNOS, COX-2, IL-1β, and TNF-α via blocking NF-κB transcription factors and MAPK signaling pathways [63]. Thus, these phenolic acids could be involved in the anti-inflammatory activity of SCE IL-1β-stimulated inflammation in ARPE-19 cells.

In summary, the present data demonstrated that when ARPE-19 cells were induced with IL-1β, cells were activated and produced inflammatory markers (IL-6, IL-8, MCP-1, iNOS, and ICAM-1) via regulation of two pathways involved in the inflammatory process. This procedure is modulated by phosphorylation at Ser536 on p65 of the NF-κB transcription factor and MAPK signaling pathways (p38, ERK1/2, and SAPK/JNK). However, when cells were pretreated with SCE and this was followed by IL-1β, SCE had an inhibitory effect on the secretion of IL-6, IL-8, and MCP-1, and the expression of iNOS and ICAM-1 in human ARPE-19 cells stimulated with IL-1β. Such action might be mediated by inhibiting the phosphorylation of MAPKs and NF-κB p65 (Figure 8).

## 4. Materials and Methods

### 4.1. Reagents

Raw sweet corn (*Zea mays* L. ssp. saccharata Sturt) was purchased from Talaad Thai, the main wholesale market (Pathum Thani Province, Thailand). Recombinant Human IL-1β (carrier-free) and enzyme-linked immunosorbent assay (ELISA) kits were purchased from BioLegend Way (San Diego, CA, USA). Specific antibodies raised against phospho-p44/42 (Erk1/2) (20G11) (#4376), p44/42 (Erk1/2) (#9102), phospho-SAPK/JNK (81E11) (#4668), SAPK/JNK (#9258), phospho-p38 (D3F9) XP^®^ (#4511), p38 (#9212), phospho-NF-kappaB p65 (Ser536) (93H1) (#3033), NF-kappaB p65 (C22B4) (#4764), iNOS (D6B6S) (#13120), CD54/ICAM-1 (E3Q9N) XP^®^ (#67836), β-actin (13E5) (#4967), and anti-rabbit IgG HRP-linked Antibody (#7074) for western blot were obtained from Cell Signaling Technology, Inc. (Danvers, MA, USA). Enhanced chemiluminescence (ECL) detection kits were acquired from Thermo Scientific (Rockford, IL, USA). Precision Plus Protein™ Kaleidoscope™ prestained protein standards were obtained from Bio-Rad Laboratories, Inc. (Hercules, CA, USA).

### 4.2. Preparation of Sweet Corn Extract (SCE)

The raw sweet corn cobs were thoroughly washed after removing the peel and corn straw. Then, the raw sweet corn cobs were boiled in reverse osmosis water for 15 min (the light yellow color of the corn grain became a bright yellow color) and cooled down in cold water for 2 min. Corn kernels (the edible portion) were sliced from the cob, ground in a kitchen blender, and lyophilized until dry. The freeze–dried samples were ground in a kitchen blender before being packed in vacuum aluminum foil and kept at −20 °C until use for the preparation of SCE.

The freeze–dried samples (1 g) were extracted with 15 mL of mixed solvent (hexane: acetone: ethanol at a 2:1:1 ratio) prior to mixing using a vortex mixer for 1 min and sonicating for 15 min (Hettich Zentrifugen, Rotina 38, North America) [39]. The sample in mixed solvent was centrifuged at 4140 g for 10 min (Hettich^®^ Instruments, Rotina 38R, Germany), and the supernatant was transferred to an amber flat-bottom flask for evaporation. The extraction procedure was repeated two times, and the clear supernatant was combined and evaporated in a rotary evaporator (Buchi Rotavapor-Re, U.S.A.) at 40 °C until dry. The dried film in the flat-bottom amber flask was completely dissolved with 5 mL mixed solvent and then transferred to an amber vial before blowing with nitrogen gas until dry. The SCE with mixed solvent was collected, weighed, and stored at −20 °C until use in the ARPE-19 cells experiments and analyzed to identify the main carotenoids of SCE using high-performance liquid chromatography method with diode array detection (HPLC–DAD). The extraction yield of SCE was 10.2 ± 0.5%.

### 4.3. Determination of Carotenoids in SCE Using HPLC–DAD

The SCE was dissolved with 2 mL of methyl tertiary-butyl ether (MtBE) and methanol (MeOH) at the ratio 20:80 and filtered through a 0.2 µm nylon syringe filter before analysis using HPLC–DAD. We used trans-β-Apo-8′-carotenal as an internal standard. Each carotenoid was identified by comparison of retention time and spectral characteristics with known standards. The concentration of carotenoid profiles in SCE was calculated by comparing the area under the curve against the standard curve with known concentrations of carotenoid standards (lutein, zeaxanthin, trans-β-Apo-8′-carotenal, β-cryptoxanthin, α-carotene, and β-Carotene) at 450 nm. Carotenoids were quantified using an HPLC system (Agilent 1260 Infinity II) with a photodiode array detector. A YMC Carotenoid column (C30) was used for separation of the carotenoids, with 2.1 mm i.d. × 150 mm length, 3 µm particle size (modified from Lee et al., 2020) [64]. The gradient mobile phase used consisted of deionized water (A), methanol (B), and methyl tertiary-butyl ether (C) (the mobile phase gradient used is shown in Table 2). The column temperature was set at 23 °C, the flow rate at 0.4 mL/min, and the detection wavelength at 450 nm (modified from Gupta et al., 2015) [65].

### 4.4. ARPE-19 Cells

Human retinal pigment epithelial (ARPE-19) cells were obtained from the American Type Culture Collection (ATCC) (CRL-2302TM, Manassas, VA 20110-2209, USA). The cells were maintained in Dulbecco’s modified Eagle medium and nutrient mixture F-12 (Ham) (1:1) (DMEM/F-12) (Life Technologies Corporation, Grand Island, NY 14072, USA). The medium was supplemented with 10% fetal bovine serum (FBS) and 1% penicillin–streptomycin. ARPE-19 cells were maintained at 37 °C in a humidified atmosphere containing 5% CO_2_ [28]. Cells were grown until more than 90% confluence prior to use in experiments. ARPE-19 cells are adherent cells and need to be trypsinized and neutralized with complete medium before centrifugation and seeding into cell culture plates for each experiment.

### 4.5. Cell Viability

Cell viability was determined using a 3-(4, 5-dimethlthiazol-2-yl)-2, 5-diphenyl tetrazolium bromide tetrazolium (MTT) assay. The MTT solution (yellow color) is converted into formazan products (purple color); this conversion is proportional to the number of live cells. Cells were washed with phosphate buffered saline (PBS) and treated with MTT solution (0.5 mg/mL of MTT in PBS) at 37 °C for 4 h. After treatment, the MTT solution was discarded, and DMSO was added to dissolve the formazan crystals produced from MTT by live cells. The absorbance was determined at 540 nm using a microplate reader (BioTek^®^ Instruments, Winooski, VT, USA).

### 4.6. Evaluation of Cytotoxicity of SCE

Cytotoxicity of SCE was determined to assess non-cytotoxic concentrations of SCE to be used in subsequent experiments. ARPE-19 cells were seeded at a density of 1 × 10^5^ cells/well in 48-well plates and then incubated for 24 h. Cells were washed with excess serum-free medium. The cells were incubated with serum-free medium that contained different SCE concentrations (1, 10, 50, 100, and 500 μg/mL) for 24 h. A concentration of 0.5% DMSO in serum-free medium was used as the control group. The viability of cells was determined using MTT assays.

### 4.7. Evaluation of a Suitable IL-β Concentration for Induction of Inflammation

ARPE-19 cells were seeded at a density of 1 × 10^5^ cells/well in 48-well plates and then maintained in medium for 24 h. Cells were washed with excess serum-free medium. The cells were incubated in serum-free medium with different IL-1β concentrations (0.1, 1, 2, 5, and 10 ng/mL) or without IL-1β (control group) for 24 h. After respective experimental treatments, the culture medium was collected to measure the expression of IL-6 and IL-8 using ELISA kits. The viability of cells was determined using MTT assays.

### 4.8. Evaluation of the Protective Effect of SCE on IL-1β-Induced Inflammation in ARPE-19 Cells

ARPE-19 cells cultured in 6-well (1.5 × 10^6^ cells/well) and 48-well (1 × 10^5^ cells/well) plates were washed with serum-free medium. Then the cells were incubated for 1 h with serum-free medium containing the maximum concentration of SCE and two more concentrations of ten-fold serial dilutions of the maximum concentration. Following 1 h incubation, the cells were treated with the appropriate IL-1β concentration for 24 h. A concentration of 0.5% DMSO in serum-free medium was used as a control group. The 48-well plate setup was used for cell viability measurements using MTT assays and the measurement of inflammatory maker concentrations (IL-6, IL-8, and MCP-1) using ELISA kits. The 6-well plate setup was used to measure inflammatory maker protein expression (iNOS, ICAM-1, MAPKs (ERK1/2, SAPK/JNK, p38 MAPK) and NF-κB p65) using western immunoblot analysis.

### 4.9. ELISA Test

The secretion of IL-6, IL-8, and MCP-1 was determined from cell culture supernatants using ELISA kits. Briefly, IL-6, IL-8, or MCP-1 capture antibodies were used to coat 96-well plates. After incubation overnight at 4 °C, plates were blocked with 1% bovine serum albumin (BSA) in PBS for 1 h. Culture medium or different concentrations of recombinant human IL-6, IL-8, and MCP-1 standard were incubated at 25 °C for 2 h. Biotinylated detection antibodies were added to the wells. The immune complexes were detected via reaction with the streptavidin–horseradish peroxidase (HRP) tetramethylbenzidine detection system. Reactions were stopped using 2 M H_2_SO_4_, and absorbance was measured using a microplate reader at 450 nm.

### 4.10. Preparation of Total Protein and Western Immunoblot Analysis

After the experimental time, cells from a 6-well plate setup were washed with cold PBS and lysed with cell lysis buffer containing phosphatase and protease inhibitors. The lysates were shaken at 4 °C for 30 min, scraped, transferred to centrifuge tubes, and centrifuged at 14,000 rpm at 4 °C for 10 min. Equal amounts of protein samples (40 µg) were loaded and separated using 10% SDS–PAGE. The proteins from each gel were transferred to 0.45 µm nitrocellulose membranes (Amersham™ Protran^®^, Sigma Aldrich, Dorset, UK). The membranes were then blocked with 5% non-fat dry milk in tris-buffered saline tween (TBST) for 1 h at 25 °C. The membranes were incubated with primary antibodies against target markers (1:1000) at 4 °C overnight. The membranes were washed with TBST and incubated with secondary antibodies against the primary antibodies (1:2000). After primary and secondary antibody incubations, protein detection was achieved using an ECL detection kit followed by imaging on an X-ray film. Band density was determined using Image J software (BioRad, Hampstead, UK). Band density values of ICAM-1, iNOS, MAPKs (phosphorylated form of ERK1/2, SAPK/JNK, p38 MAPK and total form of ERK1/2, SAPK/JNK, p38 MAPK) and NF-κB (phosphorylated form of NF-κB p65 and total form of NF-κB p65) were measured and β-actin was used to check equal loading of protein content. Results were presented as band intensity in comparison to the control group.

### 4.11. Statistical Analysis

All data in this study are expressed as mean ± SD from at least three independent experiments, and SPSS version 19 was used for statistical analyses. The differences among groups were assessed using one-way analysis of variance (ANOVA) when appropriate, followed by Duncan’s new multiple range test. Statistical significance was defined as *p* < 0.05.

## 5. Conclusions

Our in vitro data indicated a high potential of SCE to alleviate an inflammatory response, which may reduce risk factors for retinal degeneration-related disorders, especially AMD and DR. These data indicate that consumption of sweet corn may support healthy eyes. Although this study provides primary evidence to indicate the benefits of sweet corn for ocular health, further studies in animals are warranted to confirm its biological activity.

## Figures and Tables

**Figure 1 ijms-24-02462-f001:**
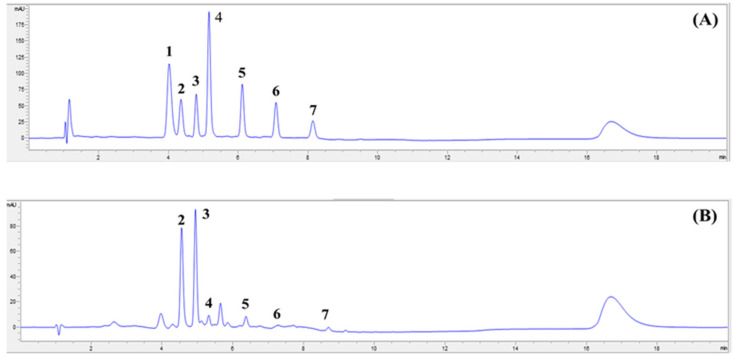
HPLC chromatogram of carotenoid standards (**A**) and sweet corn extract (SCE) (**B**) separated by HPLC–DAD at 450 nm. The compounds are (1) astaxanthin; (2) lutein; (3) zeaxanthin; (4) trans-β-Apo-8′-carotenal; (5) β-cryptoxanthin; (6) α-carotene; (7) β-carotene.

**Figure 2 ijms-24-02462-f002:**
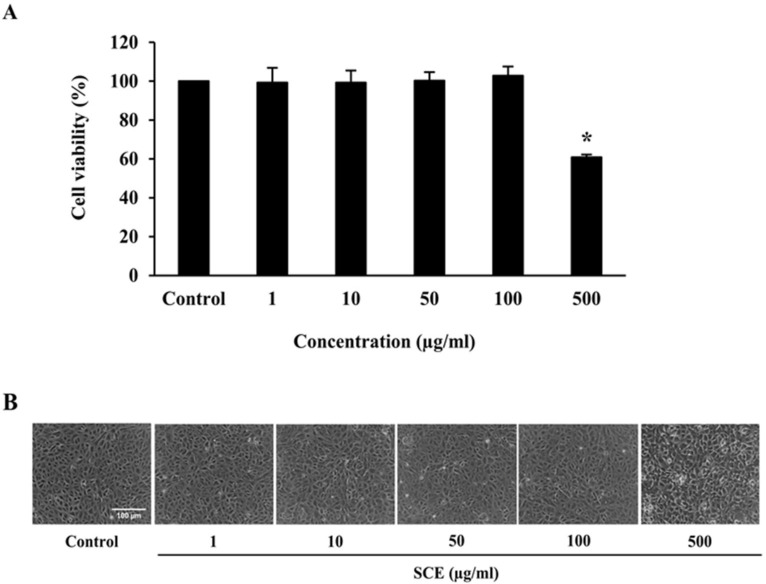
Effect of SCE on the viability of ARPE-19 cells. (**A**) ARPE-19 cells were incubated with SCE from 1 to 500 µg/mL in serum-free medium for 24 h, after which the viability of cells was evaluated using an MTT assay. A concentration of 0.5% DMSO in serum-free medium was used as the control group. Graphs represent average cell viability (mean ± SD; *n* = 3). * indicates groups are significantly different from the control group (*p* < 0.05). (**B**) Cell morphology of treated ARPE-19 was observed using phase-contrast microscopy. The scale bar represents 100 µm, 10 × magnification.

**Figure 3 ijms-24-02462-f003:**
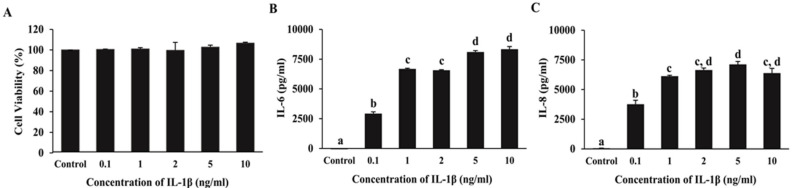
Effect of IL-1β on IL-6 and IL-8 secretion and viability of ARPE-19 cells. The viability of cells was evaluated using an MTT assay (**A**). IL-6 (**B**) and IL-8 (**C**) in the supernatants of IL-1β treated cells were measured using ELISA. Data are presented as mean ± SD (*n* = 3). The different superscripts (a–d) indicate significant differences among groups at *p* < 0.05.

**Figure 4 ijms-24-02462-f004:**
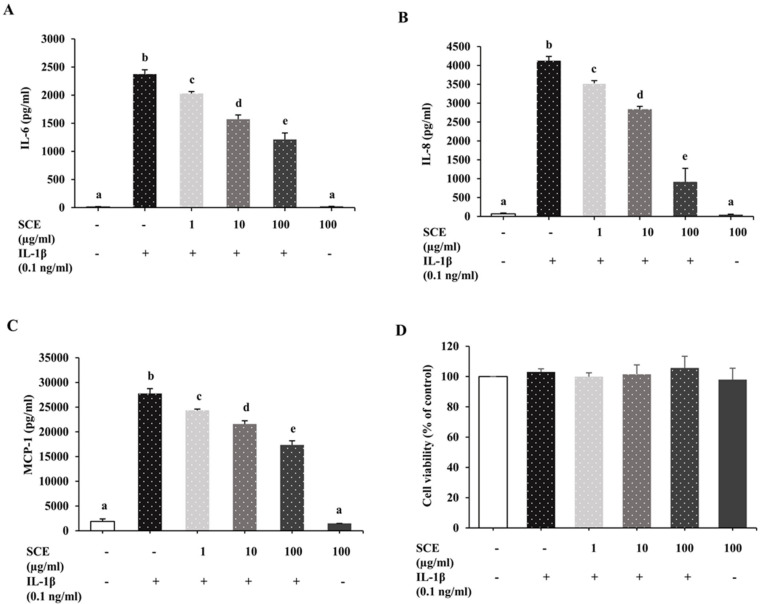
Effect of SCE on IL-6, IL-8, and MCP-1 secretion in ARPE-19 cells induced with IL-1β. ARPE-19 cells were treated with SCE at 1, 10, and 100 µg/mL in 0.5 mL of serum-free medium for 1 h before the addition of 0.1 ng/mL IL-1β for 24 h. A concentration of 0.5% DMSO in serum-free medium was used as the control group. (-) cells were not treated with SCE or IL-1β, (+) cells were treated with IL-1β. The concentrations of (**A**) IL-6, (**B**) IL-8, and (**C**) MCP-1 in culture media were measured using ELISA. (**D**) The viability of cells was assessed using an MTT assay. Data are presented as mean ± SD (*n* = 3). The different superscripts (a–e) indicate significant differences among groups at *p* < 0.05.

**Figure 5 ijms-24-02462-f005:**
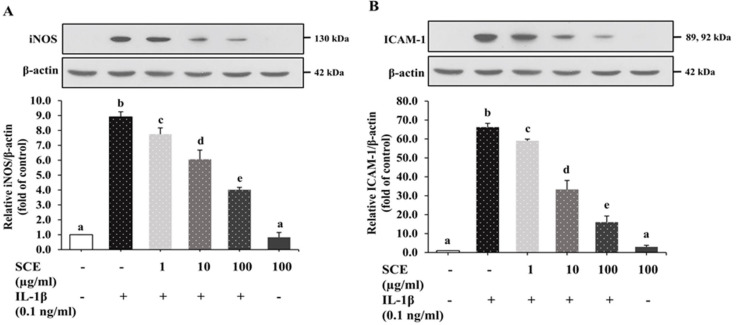
Effect of SCE on iNOS and ICAM-1 expression in IL-1β-induced ARPE-19 cells. ARPE-19 cells were treated with 1–100 µg/mL SCE for 1 h before being induced with 0.1 ng/mL IL-1β for 24 h. The protein expressions of (**A**) iNOS and (**B**) ICAM-1 were determined using immunoblotting. Band intensities of iNOS and ICAM-1 were normalized using β-actin. Data are expressed as mean ± SD (*n* = 3). The different superscripts (a–e) indicate significant differences among groups at *p* < 0.05.

**Figure 6 ijms-24-02462-f006:**
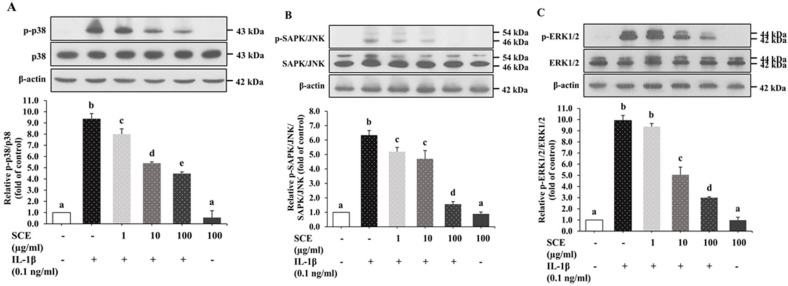
Effect of SCE on IL-1β-induced MAPK pathways in ARPE-19 cells. ARPE-19 cells were treated with 1–100 µg/mL SCE for 1 h before adding IL-1β for 24 h. Protein expressions of phosphorylated and total (**A**) p38, (**B**) SAPK/JNK, and (**C**) ERK1/2 were determined using immunoblotting of cell lysates. Band densitometry values of p-p38, p-SAPK/JNK, and p-ERK1/2 were normalized to the total form of p38, SAPK/JNK, and ERK1/2, respectively. Data are presented as mean ± SD (*n* = 3). The different superscripts (a–e) indicate significant differences among groups at *p* < 0.05.

**Figure 7 ijms-24-02462-f007:**
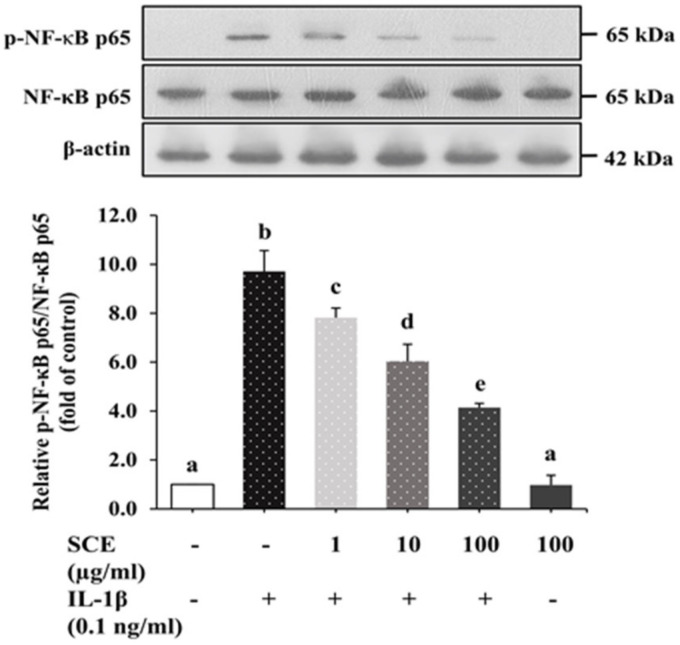
Effect of SCE on IL-1β-induced phosphorylation of NF-κB in ARPE-19 cells. ARPE-19 cells were treated with 1–100 µg/mL SCE in 0.5% DMSO for 1 h before adding IL-1β for 24 h. Protein expression of phosphorylated NF-κB p65 and total NF-κB p65 was determined using immunoblotting. Equal loading of protein in each well was normalized to the total form of NF-κB p65. Data are presented as mean ± SD (*n* = 3). The different superscripts (a–e) indicate significant differences among groups at *p* < 0.05.

**Figure 8 ijms-24-02462-f008:**
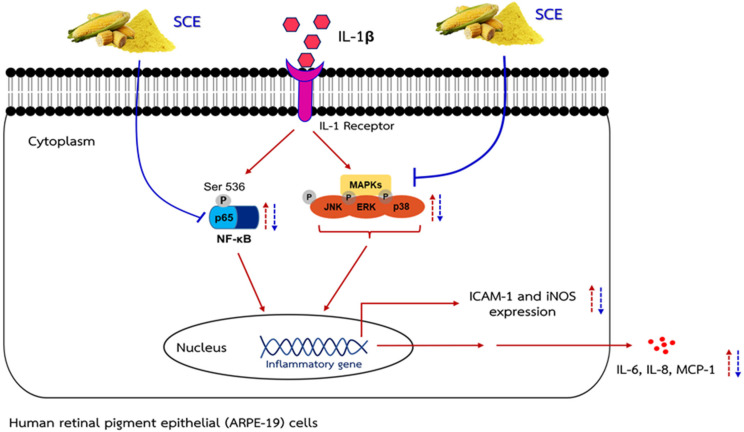
A schematic presentation of the proposed mechanism. IL-1β activated MAPK and NF-κB signaling pathways, which lead to increasing the iNOS and ICAM-1 expression and IL-6, IL-8, and MCP-1 secretion during inflammation in ARPE-19 cells. SCE treatment can decrease iNOS and ICAM-1 expression and decrease IL-6, IL-8, and MCP-1 secretion through inhibiting activation of the MAPK and NF-κB pathways.

**Table 1 ijms-24-02462-t001:** The carotenoid contents of sweet corn extract (SCE) (µg/g of crude extract).

Lutein	Zeaxanthin	β-Cryptoxanthin	α-Carotene	β-Carotene
1427.3 ± 2.3	2017.8 ± 4.6	119.9 ± 0.2	88.0 ± 1.2	126.3 ± 1.7

All values are presented as mean ± SD (*n* = 3).

**Table 2 ijms-24-02462-t002:** Mobile phase gradient used for HPLC–DAD analysis.

Time (min)	%A	%B	%C
0	4	81	15
1.5	1	69	30
5.0	0	70	30
10.0	0	0	100
13.0	0	0	100
13.1	4	81	15
20.0	4	81	15

## Data Availability

Not applicable.

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
