# Peer review of "Anti-Inflammatory Activity and Mechanism of Sweet Corn Extract on Il-1β-Induced Inflammation in a Human Retinal Pigment Epithelial Cell Line (ARPE-19)"

_ijms, 2023, doi:10.3390/ijms24032462_

Round 1

Reviewer 1 Report

I find this manuscript a good description of the work done, which supports the idea that sweet corn extracts may help reduce AMD. Nevertheless, as the authors point out, experiments with mammals are needed. I was a little concerned that experiments with individual carotenoids have already been published so the data here add little to that. Furthermore, there are many other sources of lutein and zeaxanthin---so what is special about sweet corn?

Specific comments:

Define the abbreviation SCE

75. Define MCP-1

95. Zea mays (l.c. needed)

Fig.1. What is the peak at 8.3 mins.? Label axes.

Table 1 and text. Please compare the amounts of carotenoids in sweet corn to other sources.

Fig.2(B). I could not see the scale bar.

Fig. 4. Need the vol. of media since secretion is stated as micro g/ml.

The potential benefits of phenolics in sweet corn is mentioned but the extract described is not likely to contain them because of the solvents used.

Author Response

Dear reviewer

The response to the reviewer has been attached.

Best regards

Reviewer 2 Report

This is a very well-written MS and a very well-designed study. The figures are properly designed and all data are well presented. The MS is strong.

Few minor edits are:

1. the authors are kindly asked to provide the calibration curve of the internal standard they used in the HPLC analyses,

2. figure's 4 legend needs some editing: what do (+) and (-) mean? This information needs to be added at the legend of figure 4.

3. Finally, the importance of figure 8 is substantial and it merits more discussion than the one provided with lines 433-437 in the MS. I would invite the authors to expand the description of figure 8 and explain in more detail all the biological parameters and cell molecules as they appear in this figure.

Minor revision is required.

Author Response

(The authors gave the same response as above.)
